# A novel strategy for community screening of SARS-CoV-2 (COVID-19): Sample pooling method

Khai Lone Lim[1]ᵒ, Nur Alia Johari🆔[1], Siew Tung Wong[2], Loke Tim Khaw[2], Boon Keat Tan[3], Kok Keong Chan[3], Shew Fung Wong[2,4], Wan Ling Elaine Chan[1], Nurul Hanis Ramzi[1], Patricia Kim Chooi Lim[1,2], Sulaiman Lokman Hakim[4,5], Kenny Voon🆔[2]ᵒ *

1 Institute for Research, Development and Innovation (IRDI), International Medical University, Kuala Lumpur, Malaysia, 2 Pathology Division, School of Medicine, International Medical University, Kuala Lumpur, Malaysia, 3 Human Biology Division, School of Medicine, International Medical University, Kuala Lumpur, Malaysia, 4 Centre for Environmental and Population Health Research, Institute for Research, Development and Innovation, International Medical University, Kuala Lumpur, Malaysia, 5 Department of Community Medicine, School of Medicine, International Medical University, Kuala Lumpur, Malaysia

ᵒ These authors contributed equally to this work.
* kenny_voon@imu.edu.my

**Data Availability Statement:** All relevant data are within the manuscript and its Supporting Information files

## Abstract

The rapid global spread of the coronavirus disease (COVID-19) has inflicted significant health and socioeconomic burden on affected countries. As positive cases continued to rise in Malaysia, public health laboratories experienced an overwhelming demand for COVID-19 screening. The confirmation of positive cases of COVID-19 has solely been based on the detection of the Severe Acute Respiratory Syndrome Coronavirus 2 (SARS-CoV-2) using real-time reverse transcription polymerase chain reaction (qRT-PCR). In efforts to increase the cost-effectiveness and efficiency of COVID-19 screening, we evaluated the feasibility of pooling clinical Nasopharyngeal/Oropharyngeal (NP/OP) swab specimens during nucleic acid extraction without a reduction in sensitivity of qRT-PCR. Pools of 10 specimens were extracted and subsequently tested by qRT-PCR according to the WHO-Charité protocol. We demonstrated that the sample pooling method showed no loss of sensitivity. The effectiveness of the pooled testing strategy was evaluated on both retrospective and prospective samples, and the results showed a similar detection sensitivity compared to testing individual sample alone. This study demonstrates the feasibility of using a pooled testing strategy to increase testing capacity and conserve resources, especially when there is a high demand for disease testing.

## Introduction

Coronavirus disease 2019 (COVID-19) is an infectious disease caused by a newly discovered positive-sense single-stranded RNA coronavirus known as SARS-CoV-2. It was first recognized in Wuhan, Hubei province, China, in December 2019 [1, 2]. As of 11 August 2020, a total of 19,936,210 confirmed cases of SARS-CoV-2 infection and over 732,499deaths have

**Funding:** KV, IMU477/2020, International Medical University, http://www.imu.edu.my/imu/.The funders had no role in study design, data collection and analysis, decision to publish, or preparation of the manuscript.

**Competing interests:** The authors have declared that no competing interests exist.

been reported globally [3]. In Malaysia, 9,103 cases and 125 deaths have been reported on 11 August 2020. Approximately 52% of all cases were from the densely populated Klang Valley, which comprises of the Federal Territories of Kuala Lumpur and Putrajaya as well as the state of Selangor [4]. The symptoms of SARS-CoV-2 infection are non-specific, but are usually characterized by fever, generalized weakness and dry cough [2, 5]. Most patients with COVID-19 have a mild or moderate illness and recover from the disease without needing special treatment. However, the disease can result in severe clinical manifestations and death. Thus far, the elderly or individuals with underlying medical conditions (such as chronic lung disease, diabetes, or serious heart conditions), have been observed to be more vulnerable to severe disease outcomes with COVID-19 [2].

As the COVID-19 outbreak developed into a pandemic, the WHO has recommended robust diagnostic testing to differentiate SARS-CoV-2 from other routine respiratory infections to aid in guiding appropriate clinical management. In response to the COVID-19 pandemic, Malaysia announced a nationwide movement control order (MCO) from 18 March 2020 including measures such as quarantine, isolation, social distancing, and community containment as part of efforts to curb the pandemic. Intensive contact tracing, together with increased capacity for weekly viral detection, were employed with the aim of enabling the country to resume normal day-to-day activities and operations as soon as possible. Despite efforts to increase the nationwide capacity for SARS-CoV-2 testing, including enlisting laboratory in universities, both public and private, it was overwhelmed by huge number of samples collected for testing. For example, in its daily reporting on 6 April 2020, the Crisis Preparedness and Response Centre (CPRC), Ministry of Health (MoH) infographic showed that 55,566 individuals have been tested but 8,109 results were still pending [6].

Shortage of consumables especially the RNA extraction kits because of global demand for testing, further contributed to the delay and efficiency of testing. Sample pooling, a strategy deployed for early comprehensive screening of influenza viruses and the human immunodeficiency virus (HIV) [7] and now for SARS-CoV-2 [8–11], has been demonstrated to be a cost-effective method to increase viral detection capacity in large scale diagnostic testing as well as for community screening, without compromising on the accuracy of the testing. International Medical University (IMU), a private medical university in Kuala Lumpur, responded to the call for national service by volunteering to perform the SARS-CoV-2 testing to help ease the congestion at the National Public Health Laboratory (NPHL), MoH. Here, we describe a sample pooling method for RNA extraction and qRT-PCR for the detection of SARS-CoV-2 as an alternative method to increase the cost-effectiveness of SARS-CoV-2 diagnostic tests as well as laboratories' capacity for viral detection.

As we did not perform pooled testing modelling and we were using different testing kits interchangeably, we performed a series of experiments to determine the framework for pooled testing in our laboratory. In preliminary pooled testing on clinical samples, we determined the group size for pooled testing, and followed by comparison of pooled and individual testing on clinical specimens to determine that the sensitivity of pooled testing is comparable to individual testing. We then use the validated pooled testing on the prospective clinical specimens.

## Materials and methods

### Specimen type and processing

Clinical Nasopharyngeal/Oropharyngeal (NP/OP) swab specimens used in this study were submitted to the NPHL. These specimens were from close contacts of qRT-PCR confirmed COVID-19 cases, collected in the community and primary care clinics throughout the nation. Clinical NP/OP swab specimens in viral transport media (VTM) were transported on ice to

the IMU, where all specimens were refrigerated and processed within 24 hours upon receipt of the specimens. As part of national service, IMU provided the testing service to ease the congestion at the NPHL. IMU followed the diagnostic protocol provided by NPHL and report back to NPHL the results of the testing.

## Sample pooling approach with various nucleic acid extraction kits

For this study, a total of 6 clinical specimens with a range of $C_T$ values that previously tested positive for SARS-CoV-2 by qRT-PCR was selected. Two specimens (H1-H2) demonstrated $C_T$ values in the range of 10 to 20, two (M1-M2) had $C_T$ values of 21 to 30, and another two (L1-L2) had $C_T$ values of 31 to 40. DNeasy Blood and Tissue Kit (Qiagen, Germany) and Viral Nucleic Acid Extraction Kit II (Geneaid Biotech Ltd, Taiwan) were used for nucleic acid extraction. To determine the minimum specimen volume required for the detection of SARS-CoV-2, clinical specimens of volumes 25 μL (10 specimens/pool), 40 μL (5 specimens/pool), 60 μL (10 specimens/pool) and 100 μL (5 specimens/pool) were used in the pooled samples. Nucleic acid extraction of the pooled samples was performed using both extraction kits with adjusted ratio of volume of lysis buffer to sample volume, respectively. Washing steps were performed according to the manufacturer's instructions. The extracts were eluted in 30 μL of RNase-free water and qRT-PCR was performed in triplicate for each purified nucleic acid.

## Nucleic acid extraction

Extraction of nucleic acid from clinical NP/OP swab specimens in VTM was performed using either DNeasy Blood and Tissue Kit (Qiagen, Germany) or Viral Nucleic Acid Extraction Kit II (Geneaid Biotech Ltd, Taiwan). For the extraction of individual samples, nucleic acid was extracted from 200 μL of VTM using either extraction kit according to the manufacturers' instructions. For pooled nucleic acid extraction, 10 specimens were pooled together. A volume of 60 μL of VTM from each 10 specimens was pooled together for nucleic acid extraction using either extraction kit with adjusted ratio of lysis buffer to the pooled sample volume. The washing steps were performed according to the manufacturers' instructions. During the final elution step, 30 μL of RNase-free water were used to elute the purified nucleic acid.

## Real-time reverse transcription polymerase chain reaction (qRT-PCR)

Detection of SARS-CoV-2 was performed using the qRT-PCR with primers (RdRp_SARSr-F and RdRp_SARSr-R) targeting the RNA-dependent RNA polymerase (RdRP) gene as described in the WHO-Charité protocol [12] (S1 Table). The reverse transcription and amplification reaction were performed using the SuperScript™ III One-Step RT-PCR System with Platinum™ Taq DNA Polymerase (Invitrogen, USA). Each 20 μL reaction contained 8.5 μL of RNA template, 10.0 μL of 2× reaction buffer, 0.5 μL of reverse transcriptase/Taq mixture from the kit and 1.0 μL of 10 μM RdRP-Primer/Probe mixture. All oligonucleotides were synthesized by Integrated DNA Technologies (Iowa, USA). The thermal cycling conditions used were 55˚C for 15 min for reverse transcription, followed by 95˚C for 2 min and then 45 cycles of 95˚C for 15 s and 56˚C for 30 s. A pool or sample was considered positive for COVID-19 if the $C_T$ value was less than or equal to 38.

## Preliminary pooled testing of clinical specimens

The pooled testing approach relies on the assumption that the result of the pooled sample correlates to the result of the second set of individual tests. In this case, all individuals from negative pools are considered negative for COVID-19. For the positive pools, each clinical

specimen will be re-extracted individually and tested with qRT-PCR again to determine the individual specimen(s) that are positive.

To pilot test out this strategy, 15 pooled samples (10 specimens in each pool) were carried out from 150 retrospective samples (144 negative and 6 positive clinical specimens). The 6 positive specimens were randomly added into 5 different pools. Sixty μL of VTM from each specimen were pooled together for nucleic acid extraction. Pooled specimens (600 μL per pool) were extracted with DNeasy Blood and Tissue kit (Qiagen, Germany) with adjusted 600 μL of lysis AL buffer volume and 600 μL of absolute ethanol for extraction, and the purified nucleic acid was eluted with 30 μL of RNase-free water, followed by testing using one-step real-time qRT-PCR [12]. For any pool with positive qRT-PCR results, individual specimens were extracted separately, and qRT-PCR assay was performed. The personnel performing the procedures of nucleic acid extraction and qRT-PCR were blinded from knowing which of the pools had positive samples.

### Comparison of pooled testing and individual specimen testing

To investigate the sensitivity of pooled testing and individual specimen testing, individual nucleic acid extraction for each specimen (n = 175) was performed in parallel with the 18 pooled samples (17 pools with 10 specimens each, and 1 pool with 7 specimens) prior to the qRT-PCR assay. To ensure extraction efficiency, a known positive sample was also spiked into one of the pools. PCR was performed in triplicate for each sample.

### Prospective specimens screening using sample pooling method

A total of 195 pooled samples (n = 1,745) were performed on prospective clinical NP/OP swab specimens. Each pool consists of 10 or less specimens and extracted with adjusted ratio of lysis buffer to the pooled sample volume. The subsequent procedure for nucleic acid extraction and qRT-PCR assay were performed as described above.

### Data analysis

Real-time RT-PCR results were interpreted as recommended in the WHO-Charité protocol [12]. A pool or sample was considered positive for COVID-19 if the threshold cycle ($C_T$) value was less than or equal to 38.

### Ethical statement

This study has obtained ethical clearance from the International Medical University Joint Research and Ethical Committee (EC/IRB Ref. No. 4.41/JCM-196/2020). All clinical specimens used in this study were fully anonymised and deidentified by assigning new laboratory reference number before the researchers can access them. The researchers were blinded from the patients' information.

## Results

### Comparison of different volumes for sample pooling

While maintaining the sensitivity of qRT-PCR detection of SARS-CoV-2, we investigated the minimal volume of specimen required for the assay. Clinical specimens of different volumes, 25 μL (10 specimens/pool), 40 μL (5 specimens/pool), 60 μL (10 specimens/pool) and 100 μL (5 specimens/pool) were used. One positive specimen was included in each pool and mixed with negative specimens. We used positive specimens with different range of $C_T$ values, H1 and H2 ($C_T$ values 10–20), M1 and M2 ($C_T$ values 20–30), and L1 and L2 ($C_T$ values 30–38) to

ensure that the pooled testing does not reduce the sensitivity of SARS-CoV-2 detection by qRT-PCR for clinical specimens with low viral loads.

Positive signals were detected in pooled samples using 25 µL, 40 µL, 60 µL and 100 µL of specimen volume for H1, H2, M1 and M2 (Table 1). However, such a trend was not observed for pooled samples using L1 and L2. Pooled samples with 60 µL and 100 µL of L1 and L2 showed positive signals while those with 25 µL and 40 µL of specimen volume resulted in negative signals. We observed that $C_T$ values of positive and negative signals from positive and negative samples, respectively, did not vary between the nucleic acid extracted with either the Qiagen DNeasy Blood and Tissue Kit and Geneaid Viral Nucleic Acid Extraction Kit II (Tables 1 and S2).

## Proof of concept with retrospective samples

Based on Table 1, 60 µL of VTM were chosen to pilot test the concept of pooling strategy. Clinical specimens previously established as positive for the *RdRP* gene target of SARS-CoV-2 were chosen to determine whether they were detectable when mixed with negative specimens. Pools of 10 clinical specimens were examined, with four pools containing one positive and nine negative specimens. One pool contained two positives with eight negative specimens. Positive

**Table 1. Effect of clinical specimen volume in pooled samples for the detection of Severe Acute Respiratory Syndrome Coronavirus 2 (SARS-CoV-2) by qRT-PCR.**

| Specimen ID | Volume of specimen used (µL) | $C_T$ value[a] | |
|---|---|---|---|
| | | Qiagen DNeasy Blood and Tissue Kit | Geneaid Viral Nucleic Acid Extraction Kit II |
| H1 (IMU0094) | 100[b] | 18.35 (0.41) | 18.25 (0.19) |
| | 60[c] | 21.21 (0.30) | 22.41 (0.54) |
| | 40[b] | 21.85 (0.33) | 22.33 (1.02) |
| | 25[c] | 22.08 (0.56) | 22.51 (0.68) |
| H2 (IMU0093) | 100[b] | 18.61 (0.32) | 19.25 (0.27) |
| | 60[c] | 20.88 (0.21) | 20.78 (0.40) |
| | 40[b] | 22.48 (0.44) | 22.61 (0.59) |
| | 25[c] | 23.43 (1.01) | 24.22 (1.20) |
| M1 (IMU0047) | 100[b] | 22.85 (1.02) | 22.83 (0.45) |
| | 60[c] | 25.34 (0.31) | 24.64 (0.32) |
| | 40[b] | 25.87 (0.18) | 24.93 (0.67) |
| | 25[c] | 25.82 (0.52) | 26.07 (0.24) |
| M2 (IMU0310) | 100[b] | 27.73 (0.94) | 28.41 (1.09) |
| | 60[c] | 28.40 (0.46) | 28.45 (0.46) |
| | 40[b] | 29.20 (0.30) | 29.07 (0.78) |
| | 25[c] | 30.44 (0.60) | 30.21 (1.13) |
| L1 (IMU0254) | 100[b] | 31.58 (0.44) | 31.70 (1.02) |
| | 60[c] | 32.25 (0.41) | 34.24 (1.16) |
| | 40[b] | 33.92 (0.66) | Not Detected |
| | 25[c] | Not Detected | Not Detected |
| L2 (IMU0255) | 100[b] | 35.40 (0.16) | 34.95 (0.92) |
| | 60[c] | 35.74 (0.39) | 36.11 (0.16) |
| | 40[b] | Not Detected | Not Detected |
| | 25[c] | Not Detected | Not Detected |

[a] The $C_T$ values are expressed as mean (standard deviation); A pool or sample was considered positive for COVID-19 if the $C_T$ value was less than or equal to 38.

[b] The volume of specimen used in 5-sample pool (1 positive + 4 negatives).

[c] The volume of specimen used in 10-sample pool (1 positive + 9 negatives).

specimens with a different range of $C_T$ values were chosen, and PCR was performed in triplicate for determination of reproducibility. The mean $C_T$ value of the pooled specimen was compared to the mean $C_T$ value of the positive specimen diluted in VTM. The results showed differences of one to two $C_T$ demonstrating the effectiveness of pooling 10 specimens (Table 2). We also noted positive $C_T$ values on amplification of pooled samples for groups H and I even though no $C_T$ values were detected on the first and second time qRT-PCR on individual samples from these two groups.

## Comparison of individual and pooled testing in parallel

To address whether inter-performer variation may be one of the factors contributing to detectable viral load sample pooling, individual sampling and pooled sampling were performed in parallel. As the maximum testing capacity for the laboratory was 100 individual specimens per day, the individual versus pooled testing experiment was performed over a period of two days. One positive sample was implanted among each sample group (N = 89 + 86, these numbers represent tests performed over two separate days) that was collected prospectively. This study demonstrated that pooled sampling detection was comparable to individual sampling (S3 Table). Positive signals that were detected among the pools of positive samples were identified to be the implanted positive samples. It was noted that pooled negative samples did not result in a detectable $C_T$ value among the prospective samples.

**Table 2. Preliminary testing of sample pooling method on retrospective samples for the detection of Severe Acute Respiratory Syndrome Coronavirus 2 (SARS-CoV-2) by qRT-PCR.**

| Pool ID | Pooled samples (10 specimens/pool)[b] | Results of qRT-PCR[a] ($C_T$ value) | |
|---|---|---|---|
| | | **Individual testing** | **Pooled testing[c]** |
| A | 1 positive (IMU0255) + 9 negatives | $C_T$ = 36.24 (1.35) | $C_T$ = 37.54 (0.40) |
| B | 10 negatives | Not Detected | Not Detected |
| C | 10 negatives | Not Detected | Not Detected |
| D | 10 negatives | Not Detected | Not Detected |
| E | 1 positive (IMU0094) + 9 negatives | $C_T$ = 17.59 (0.90) | $C_T$ = 22.36 (0.22) |
| F | 10 negatives | Not Detected | Not Detected |
| G | 2 positives (IMU0254 & IMU0256) + 8 negatives | $C_T$ = 31.24 (0.36) | $C_T$ = 36.40 (1.04) |
| | | $C_T$ = 37.49 (0.49) | |
| H* | 10 negatives | Not Detected | $C_T$ = 39.46 (0.57) |
| I* | 10 negatives | Not Detected | $C_T$ = 40.38 (0.60) |
| J | 1 positive (IMU0448) + 9 negatives | $C_T$ = 30.02 (0.33) | $C_T$ = 31.39 (0.29) |
| AA | 10 negatives | Not Detected | Not Detected |
| BB | 10 negatives | Not Detected | Not Detected |
| CD | 10 negatives | Not Detected | Not Detected |
| DD | 10 negatives | Not Detected | Not Detected |
| EE | 1 positive (IMU0139) + 9 negatives | $C_T$ = 37.06 (0.32) | $C_T$ = 37.70 (0.17) |

[a] The qRT-PCR was performed in triplicates; The $C_T$ values are expressed as mean (standard deviation); A pool or sample was considered positive for COVID-19 if the $C_T$ value was less than or equal to 38.

[b] All clinical NP/OP swab specimens used were previously tested for SARS-CoV-2 (n = 150); Six clinical specimens with a range of $C_T$ values that previously tested positive for SARS-CoV-2 by qRT-PCR assay were used to spike the pools.

[c] The volume of clinical specimen used for pooled testing (10-sample pool) is 60 μL.

* $C_T$ values of 39.46 and 40.38 were detected for Groups H and I, respectively. These results were regarded as negative for COVID-19 in this case.

## Sampling pooling—the way onwards

A total of 195 pools composed of 1745 individual specimens were screened (Fig 1), and 36 positive specimens were identified. Thirty-four positive specimens were identified through pooled screening on Day 6 (April 18, 2020) belonging to a new COVID-19 cluster of students returning to Malaysia from Indonesia on April 16, 2020 [13, 14]. Clinical NP/OP swab specimens were collected in VTM on April 17, 2020 and sent to the laboratory on April 18, 2020 for the screening of COVID-19.

## Discussion

The detection of COVID-19 using qRT-PCR with the pooling of samples during nucleic acid extraction greatly reduced workload and costs especially when disease prevalence was low [15, 16]. In the present study, the cost reduction of RNA extraction without reduction of qRT-PCR sensitivity was achieved by pooling 10 specimens for the initial qRT-PCR screening as described in previous work involving the estimation of HIV and Influenza virus prevalence [16, 17]. The pooling of samples for COVID-19 qRT-PCR was also evaluated by Yelin et al. (2020) [11]. Similarly, in this study, pooling was performed on clinical NP/OP swab specimens in VTM, thus allowing a reduction in the number of extractions and qRT-PCRs performed by at least nine-fold. The determination of optimal pool size for each geographical area depends not only on expected prevalence but also on resources available for testing, number of samples available and objective of the testing strategy. When the prevalence is low, such as in Malaysia where the majority of the sample cohort positivity rate was less than 5%, a large proportion of the samples are expected to be negative. Thus, a pooling strategy like this will significantly

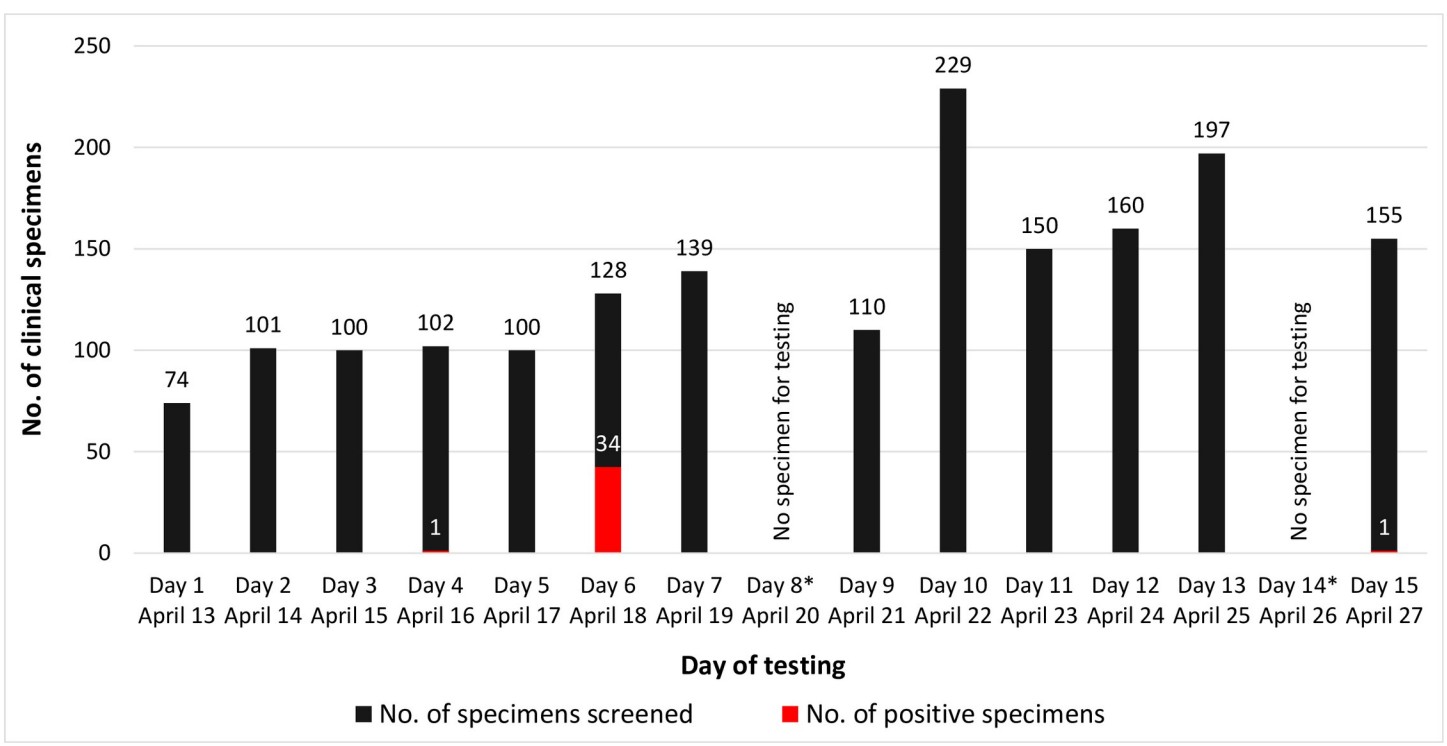

**Fig 1. Number of clinical specimens tested for Severe Acute Respiratory Syndrome Coronavirus 2 (SARS-CoV-2).** Testing was performed by pooled sample screening at the IMU Advanced Microbiology Collaborative Research Laboratory since April 13, 2020 (S1 Dataset). Each pool included 10 clinical NP/OP swab specimens. A total of 195 pools comprising of 1,745 clinical specimens was screened; * There was no specimen received for testing on Day 8 (April 20, 2020) and Day 14 (April 26, 2020).

improve efficiency and will help to reduce the backlog of pending COVID-19 tests, and subsequently reduce diagnostic turn-around-time which is critical for patient management and transmission control (isolation and quarantine measures). It will be useful for resource-strained laboratories especially in developing countries. With this approach, testing may be more feasible and affordable if the government chooses to screen samples from larger communities.

During the Movement Control Order (MCO), Malaysia faced a shortage of both viral RNA extraction and qRT-PCR kits. In addition, with more than 8,000 results still pending during the MCO period, pooled testing was necessarily required. Pooled testing for COVID-19 has been practiced in several laboratories from different regions [8–11], with most laboratories practicing pooled testing for qRT-PCR. Here we describe the pooled testing of 10 specimens starting from viral RNA extraction and followed by qRT-PCR while maintaining the reliability of testing. Although we did not perform any modelling on pool size prior to testing, we observed a threshold cycle ($C_T$ value) difference of less than 3 between individual and pooled testing (S2 Table), and these findings are consistent with recent studies [11]. Nevertheless, a previous study conducted by Abdalhamid et al. (2020) has shown that determination of optimal pool size prior to testing was capable of improving the overall efficiency of pooled testing [10]. Based on the web-based application for pooling, a pool size with 5 specimens may conserve more resources than a pool size of 10 as the prevalence rate increases.

Noticeable $C_T$ values were generated in pools of negative samples (H and I) in retrospective samples (Table 2) at the first round of qRT-PCR. Repeated individual extractions demonstrated that the samples were negative. These results were regarded as negative for COVID-19 in this case. Similar findings were also reported in a recent study based in California, the United States, where one of 292 pools showed a positive E gene signal when screened using qRT-PCR but tested negative when samples from that pool were screened individually [8].

Our results suggest that the sample pooling method established herein is a robust and cost-effective strategy for the detection of SARS-CoV-2 in pools of 10 clinical NP/OP swab specimens. Furthermore, two different nucleic acid extraction kits, namely the Qiagen DNeasy Blood and Tissue Kit and Geneaid Viral Nucleic Acid Extraction Kit II, were used in this study interchangeably as we were facing resource limitations during the MCO period in Malaysia. Even though we did not compare the sensitivity and specificity of the two extraction kits, both extraction kits showed consistent results for the detection of positive and negative signals from positive and negative samples, respectively (S2 Table).

Our finding is important as it has demonstrated that the nucleic acid extraction kits are interchangeable while the sensitivity remained high for COVID-19 diagnostic test. Till today, many countries are still facing a shortage of COVID-19 testing kits. It is safe to believe that not every laboratory has unlimited supplies of the same testing kits throughout the testing period. Thus, improvisation of protocols may be necessary to overcome such a shortcoming. In general, the use of suggested pooled testing may require validation for effective COVID-19 diagnostic tests as protocols vary from laboratory to laboratory and manufacturer to manufacturer.

The pooled testing strategy used in this study does not require significant structural or workflow changes for laboratories that are currently performing the screening of SARS-CoV-2 by qRT-PCR. However, laboratories are recommended to develop and validate the workflow scheme in-house for the tracking of specimens in pools. This study has a few limitations. Firstly, a reference strain of SARS-CoV-2 and samples that were positive for other respiratory viruses were not included in this study to test for potential inhibitory effects of cellular material from pooling multiple clinical specimens and/or cross-reactivity. Secondly, the pooling approach works well when the disease positivity rate is low. The benefit of pooling, particularly with respect to the conservation of reagents, is nullified if every pool yields a positive result.

Thus far (as of 27 April 2020) our laboratory detected 52 positives among the 2,732 samples collected through contract tracing; i.e. a prevalence rate of 1.90%. The pooling approach worked spectacularly well when the prevalence is low. However, as prevalence increased, 5 specimens in a pool may be more practical than 10 specimens. Our data showed that 5-sample pooling versus 10-sample pooling did not show any significant difference in $C_T$ value, and thus we chose 10-sample pooling based on the low prevalence rate of positive COVID-19 in our specimens. A threshold cycle ($C_T$ value) difference of less than 5 was observed between 5-sample and 10-sample pooling as shown in Table 1. Overall, this study has shown that the pooled screening of COVID-19 has increased overall testing capacity of the COVID-19 with limited resources. Through pooled screening strategy, we were able to increase initial testing capacity, from 100 to 500 specimens in a day.

In summary, we have demonstrated that clinical NP/OP swab specimens in VTM can be pooled and tested for the presence of SARS-CoV-2 without sacrificing sensitivity. Pooled testing is a resource-efficient strategy for the detection of early community transmission of COVID-19.

## Supporting information

**S1 Table. Primers and probes for qRT-PCR of SARS-CoV-2.**
(PDF)

**S2 Table. Comparison of individual and pooled testing in different nucleic acid extraction kits[a].**
(PDF)

**S3 Table. Comparison of individual and pooled testing[b].**
(PDF)

**S1 Dataset. IMU COVID-19 pooled testing results (April 13–27, 2020).**
(PDF)

## Acknowledgments

We thank the National Public Health Laboratory (NPHL), MoH, Malaysia for their generosity and support in sharing the necessary reagents for laboratory testing.

## Author Contributions

**Conceptualization:** Khai Lone Lim, Sulaiman Lokman Hakim, Kenny Voon.

**Data curation:** Khai Lone Lim.

**Formal analysis:** Khai Lone Lim, Patricia Kim Chooi Lim.

**Funding acquisition:** Sulaiman Lokman Hakim, Kenny Voon.

**Investigation:** Khai Lone Lim, Nur Alia Johari, Siew Tung Wong, Loke Tim Khaw, Boon Keat Tan, Kok Keong Chan, Shew Fung Wong, Wan Ling Elaine Chan, Nurul Hanis Ramzi, Kenny Voon.

**Methodology:** Khai Lone Lim, Patricia Kim Chooi Lim, Sulaiman Lokman Hakim, Kenny Voon.

**Validation:** Khai Lone Lim.

**Writing – original draft:** Khai Lone Lim, Nur Alia Johari, Patricia Kim Chooi Lim, Kenny Voon.

**Writing – review & editing:** Khai Lone Lim, Nur Alia Johari, Siew Tung Wong, Loke Tim
Khaw, Boon Keat Tan, Kok Keong Chan, Shew Fung Wong, Wan Ling Elaine Chan, Nurul
Hanis Ramzi, Patricia Kim Chooi Lim, Sulaiman Lokman Hakim, Kenny Voon.

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
