## [Decision Letter · Decision Letter 0]

25 Jun 2020

PONE-D-20-12301

A novel strategy for community testing of SARS-CoV-2 (COVID-19): sample pooling method

PLOS ONE

Dear Dr. Voon,

Thank you for submitting your manuscript to PLOS ONE. After careful consideration, we feel that it has merit but does not fully meet PLOS ONE’s publication criteria as it currently stands. Therefore, we invite you to submit a revised version of the manuscript that addresses the points raised during the review process.

I have received the review of your manuscript. While your paper addresses an interesting question, the reviewer stated several concerns about your study that need to be address carefully.  Please see reviewer’s insightful comments below.  Personally, at a more detailed level, I am not convinced the argument made by the authors that pooling negative samples may generate false positives by increasing the viral load to a detectable limit (line 199 – 204 and line 271 – 280). I am more inclined to think the opposite may be more true, pooled samples tend to dilute the viral load. I think different explanation will need to be made.  Additionally, in line 184, the authors referred to S1 Table, but the information is also available in table 1. Suggest including both Table 1 and S1 Table here.

We look forward to receiving your revised manuscript.

Kind regards,

Baochuan Lin, Ph.D.

Academic Editor

PLOS ONE

Journal Requirements:

2. In your methods section, please disclose the sequences of the RT-PCR primers and ensure that their original source is cited, if applicable. Also, please briefly describe the recommendations of the WHO-Charité protocol in relation to the interpretation of results.

3. In your ethics statement and methods section, please clarify whether the biological samples used in your study were de-identified and fully anonymized before the researchers accessed them. If not, please clarify whether patients provided consent to use their samples or whether the ethics committee waived the need for consent.

Reviewers' comments:

Reviewer's Responses to Questions

**Comments to the Author**

1. Is the manuscript technically sound, and do the data support the conclusions?

Reviewer #1: Yes

2. Has the statistical analysis been performed appropriately and rigorously? 

Reviewer #1: Yes

3. Have the authors made all data underlying the findings in their manuscript fully available?

Reviewer #1: No

4. Is the manuscript presented in an intelligible fashion and written in standard English?

Reviewer #1: Yes

5. Review Comments to the Author

Reviewer #1: The purpose of this paper is to develop a specimen pooling procedure that can be used for SARS-CoV-2 detection in samples. The procedure is validated with known positive specimens and also applied in clinical practice. Overall, I think this is an important contribution and should be published subject to minor revisions.

My comments about the paper are as follows.

1) “Pooling” goes by a few other different names: pooled testing and group testing. It would be good to include these names as key words or include somewhere else in the paper so that search engines can find this paper.

2) The application of pooling to test for SARS-CoV-2 has many papers already, where most are available on pre-print servers like MedRxiv and ArXiv. The only peer-reviewed papers that I am aware of that use actual specimens are

• Abdalhamid et al. (American Journal of Clinical Pathology, 2020), https://academic.oup.com/ajcp/article/153/6/715/5822023

• Hogan et al. (Journal of the American Medical Association, 2020), https://jamanetwork.com/journals/jama/fullarticle/2764364

• Lohse et al. (Lancet Infectious Diseases, 2020), https://www.thelancet.com/journals/laninf/article/PIIS1473-3099(20)30362-5/fulltext

• Yelin et al. (Clinical Infectious Diseases, 2020), https://academic.oup.com/cid/advance-article/doi/10.1093/cid/ciaa531/5828059

The Yelin et al. (2020) paper is already mentioned and differences between it and the submitted paper are described. The paper is now published, so its reference should be updated. The Hogan et al. (2020) paper is mentioned as well, but differences between it and the submitted paper need to be described. For the other two papers, you should include them, while also describing how their work differs with the submitted paper.

3) There are a number of separate investigations performed in the paper, and it was difficult to keep track of them all through initial readings. Overall, I see three main investigations:

a) “Preliminary testing of sample pooling method”,

b) “Comparison of sampling pooling and individual specimen testing”, and

c) “Prospective specimens screening using sample pooling method”

In the last paragraph of the introduction, one sentence could be given for each to explain their purpose and what is being done. This will help foreshadow to the reader what to expect in the remainder of the paper. Better linkages between these sections may be helpful too.

4) Page 9 discusses the different volumes used. While it provides results for M2 and L2, what about the results for H1-H2, M1, and L2?

5) I am concerned about how standard deviations are represented in Tables 1 and 2 by “mean ± standard deviation”. While I have seen this representation in some other publications before, it is actually misleading because a reader may think all values are within a range of mean – standard deviation and mean + standard deviation or a reader may think this is some type of confidence interval. Better approaches for including standard deviations include “mean (standard deviation)” with a table caption or column header stating something like “mean (standard deviation) of …” or a separate column in the table for the standard deviations.

6) Page 11 states “n = 89 + 86” but it is not clear until page 15 why these numbers are given separately like this. I recommend giving a brief explanation on page 11 that these numbers represent tests performed over two separate days.

7) In the Discussion section, I really liked seeing comments about prevalence and how this can be used to determine an appropriate group size. However, I think more guidance should be given about selecting group size. The statistical literature has many papers on pooling where guidance could be obtained. For example, these papers include Hitt et al. (Statistics in Medicine, 2019) and Kim et al. (Biometrics, 2007). The Abdalhamid et al. (2020) is the only paper on SARS-CoV-2 detection that I am familiar which addresses group size selection while also using actual samples.

8) The data corresponding to Figure 1 should be made available, perhaps in a format like what is given in Table 3 (each pool is displayed with the positive and negative indicators). Rather than an actual table within the paper, I think a data file would be sufficient. One reason to include the “raw” data rather than only the summarized data as in Figure 1 is that it allows readers to see how often multiple positives occur in the same pool. A summary perhaps could provide this same level of detail with

Number of positives Number of pools

0 1709

1 ___

2 ___

…

10 ___

rather than a data file or table like Table 3. One aspect that would be missing though is the time element. Given only three days of positives, this may not be too much of a loss.

6. PLOS authors have the option to publish the peer review history of their article (what does this mean?). If published, this will include your full peer review and any attached files.

Reviewer #1: No

---

## [Author Response · Author response to Decision Letter 0]

17 Jul 2020

We thank the reviewers for their generous comments on the manuscript and have edited the manuscript to address their concerns. Please find attached the detailed responses to the reviewer’s comments and the revised version of the manuscript. We have made all necessary amendments and modifications following the reviewers’ recommendations.

We truly value and appreciate your opinions and helpful contents in improving the manuscript.

---

## [Decision Letter · Decision Letter 1]

11 Aug 2020

PONE-D-20-12301R1

A novel strategy for community screening of SARS-CoV-2 (COVID-19): Sample pooling method

PLOS ONE

Dear Dr. Voon,

Thank you for submitting your manuscript to PLOS ONE. After careful consideration, we feel that it has merit but does not fully meet PLOS ONE’s publication criteria as it currently stands. Therefore, we invite you to submit a revised version of the manuscript that addresses the points raised during the review process.

The revised version has showed significant improvement, however, there are still a few issues that need to be addressed (please see specific comments and reviewer's comments below).

Specific comments:

1. Line 100, "...transported in cold to..." Did the authors mean shipped in ice or dry ice? Transported in cold is very vague, please clarify.

2. Table 3 can be supplemental table.

We look forward to receiving your revised manuscript.

Kind regards,

Baochuan Lin, Ph.D.

Academic Editor

PLOS ONE

Reviewers' comments:

Reviewer's Responses to Questions

**Comments to the Author**

1. If the authors have adequately addressed your comments raised in a previous round of review and you feel that this manuscript is now acceptable for publication, you may indicate that here to bypass the “Comments to the Author” section, enter your conflict of interest statement in the “Confidential to Editor” section, and submit your "Accept" recommendation.

Reviewer #1: (No Response)

2. Is the manuscript technically sound, and do the data support the conclusions?

Reviewer #1: Yes

3. Has the statistical analysis been performed appropriately and rigorously? 

Reviewer #1: Yes

4. Have the authors made all data underlying the findings in their manuscript fully available?

Reviewer #1: Yes

5. Is the manuscript presented in an intelligible fashion and written in standard English?

Reviewer #1: Yes

6. Review Comments to the Author

Reviewer #1: I have only a few additional comments. All line references are with respect to the tracked changes version of the paper.

1) There are a number of references to items from April that could be updated. For example, the number of confirmed cases of SARS-CoV-2 stated was for April 22.

2) Lines 205-208: Could sigmoidal curves for L1 and L2 be examined to see if potentially more cycles could have been useful? This approach was taken by Yelin et al. (2020).

3) Lines 295:298: Yelin et al. (2020) also looked at pooling before extraction (see “Pooling Prior to RNA Extraction” subsections). Please change your wording in the discussion section.

4) Lines 318-320 and reply to referee comments: While modeling for pool sizes prior to implementation was not performed, a reference should be made showing how this could be done. This is an important aspect of pooling to make sure it can be done as efficiently as possible. After all, pooling is being used for efficiency purposes. For example, if a pool size of 5 led to much fewer tests than a pool size of 10, wouldn’t a pool size of 5 be better to use then?

7. PLOS authors have the option to publish the peer review history of their article (what does this mean?). If published, this will include your full peer review and any attached files.

Reviewer #1: No

---

## [Author Response · Author response to Decision Letter 1]

12 Aug 2020

We would like express our appreciation to the reviewers for their feedback on the manuscript and have edited the manuscript to address their concerns. Please find attached the detailed responses to the reviewer’s comments and the revised version of the manuscript. We have made all necessary amendments and modifications following the reviewers’ recommendations.

---

## [Decision Letter · Decision Letter 2]

18 Aug 2020

A novel strategy for community screening of SARS-CoV-2 (COVID-19): Sample pooling method

PONE-D-20-12301R2

Dear Dr. Voon,

We’re pleased to inform you that your manuscript has been judged scientifically suitable for publication and will be formally accepted for publication once it meets all outstanding technical requirements.

Kind regards,

Baochuan Lin, Ph.D.

Academic Editor

PLOS ONE

Additional Editor Comments (optional):

Reviewers' comments:

Reviewer's Responses to Questions

**Comments to the Author**

1. If the authors have adequately addressed your comments raised in a previous round of review and you feel that this manuscript is now acceptable for publication, you may indicate that here to bypass the “Comments to the Author” section, enter your conflict of interest statement in the “Confidential to Editor” section, and submit your "Accept" recommendation.

Reviewer #1: All comments have been addressed

2. Is the manuscript technically sound, and do the data support the conclusions?

Reviewer #1: (No Response)

3. Has the statistical analysis been performed appropriately and rigorously? 

Reviewer #1: (No Response)

4. Have the authors made all data underlying the findings in their manuscript fully available?

Reviewer #1: (No Response)

5. Is the manuscript presented in an intelligible fashion and written in standard English?

Reviewer #1: (No Response)

6. Review Comments to the Author

Reviewer #1: (No Response)

7. PLOS authors have the option to publish the peer review history of their article (what does this mean?). If published, this will include your full peer review and any attached files.

Reviewer #1: No

---

## [Editor Report · Acceptance letter]

20 Aug 2020

PONE-D-20-12301R2 

A novel strategy for community screening of SARS-CoV-2 (COVID-19): sample pooling method 

Dear Dr. Voon:

I'm pleased to inform you that your manuscript has been deemed suitable for publication in PLOS ONE. Congratulations! Your manuscript is now with our production department. 

Kind regards, 

on behalf of

Dr. Baochuan Lin 

Academic Editor

PLOS ONE